# Visual Acuity Gain Profiles and Anatomical Prognosis Factors in Patients with Drug-Naive Diabetic Macular Edema Treated with Dexamethasone Implant: The NAVEDEX Study

**DOI:** 10.3390/pharmaceutics13020194

**Published:** 2021-02-01

**Authors:** Mauricio Pinto, Thibaud Mathis, Pascale Massin, Jad Akesbi, Théo Lereuil, Nicolas Voirin, Frédéric Matonti, Franck Fajnkuchen, John Conrath, Solange Milazzo, Jean-François Korobelnik, Stéphanie Baillif, Philippe Denis, Catherine Creuzot-Garcher, Mayer Srour, Bénédicte Dupas, Aditya Sudhalkar, Alper Bilgic, Ramin Tadayoni, Eric H Souied, Corinne Dot, Laurent Kodjikian

**Affiliations:** 1Department of Ophthalmology, Croix-Rousse University Hospital, 69004 Lyon, France; mauricio.pinto@chu-lyon.fr (M.P.); mathisthibaud@hotmail.fr (T.M.); theo.lereuil@chu-lyon.fr (T.L.); philippe.denis@chu-lyon.fr (P.D.); 2UMR-CNRS 5510 Matéis, 69100 Villeurbane, France; 3Department of Ophthalmology, Lariboisière Hospital (AP-HP), University Paris 7 (Sorbonne Paris Cité), 75007 Paris, France; pascale.massin1@gmail.com (P.M.); benedicte.dupas@gmail.com (B.D.); sec.tadayoni@gmail.com (R.T.); 4Quinze-Vingt National Eye Hospital, 75012 Paris, France; akesbi@gmail.com; 5EPIMOD, Epidemiology and Modelling, 01240 Dompierre sur Veyle, France; nivoirin@gmail.com; 6Department of Ophthalmology, Marseille North Hospital, 13015 Marseille, France; fredmatonti@gmail.com; 7Department of Ophthalmology, Monticelli’s Clinic, 13008 Marseille, France; johnconrath@yahoo.fr; 8Department of Ophthalmology, Avicenne Hospital, 93022 Bobigny, France; frank.fajnkuchen@gmail.com; 9Department of Ophthalmology, University Hospital of Amiens-Picardie, 80021 Amiens, France; dr.solange.milazzo@gmail.com; 10Department of Ophtalmology, Bordeaux University Hospital, 33000 Bordeaux, France; jean-francois.korobelnik@chu-bordeaux.fr; 11Bordeaux Population Health Research Center, Team LEHA, UMR 1219, INSERM, University Bordeaux, 33000 Bordeaux, France; 12Department of Ophthalmology, University Hospital Pasteur 2, 06000 Nice, France; baillifsteph@yahoo.fr; 13Department of Ophthalmology, University Hospital François Mitterrand, 21000 Dijon, France; catherine.creuzot-garcher@chu-dijon.fr; 14Department of Ophthalmology, Creteil Intercommunal Hospital, University Paris Est Creteil, 94000 Creteil, France; srour.mayer@gmail.com (M.S.); esouied@hotmail.com (E.H.S.); 15Alphavision Augenzentrum, 27570 Bremerhaven, Germany; adityasudhalkar@icloud.com (A.S.); drbilgicalper@yahoo.com (A.B.); 16MS Sudhalkar Medical Research Foundation, Baroda 390001, India; 17Department of Ophthalmology, Desgenettes Military Hospital, 69003 Lyon, France; corinnedot.pro@hotmail.fr; 18French Military Health Service Academy of Val-de-Grâce, 75005 Paris, France

**Keywords:** dexamethasone-implant, diabetic macular edema, visual acuity gain

## Abstract

**Brief Summary Statement:**

NAVEDEX (NAive diabetic macular Edema treated by DEXamethasone implant) study is a real-life multi-center study on drug-naive diabetic macular edema treated by Dexamethasone-implant. Two different visual acuity gain (VA) profiles were identified, based on baseline visual acuity (VA). Baseline disorganization of retinal inner layers or ellipsoid zone alterations (EZAs) negatively influence final VA but has no impact on VA gain.

**Abstract:**

The purpose of this study is to evaluate the visual acuity (VA) gain profiles between patients with drug-naive diabetic macular edema (DME) treated by dexamethasone implant (DEX-implant) and assess the baseline anatomical and functional factors that could influence the response to the treatment in real-life conditions. A retrospective, multi-center observational study included 129 eyes with drug-naive DME treated by DEX-implant. The Median follow-up was 13 months. Two groups of VA gain trajectories were identified—Group A, with 71% (*n* = 96) of patients whose average VA gain was less than five letters and Group B, with 29% (*n* = 33) of patients with an average gain of 20 letters. The probability of belonging to Group B was significantly higher in patients with baseline VA < 37 letters (*p* = 0.001). Ellipsoid zone alterations (EZAs) or disorganization of retinal inner layers (DRILs) were associated with a lower final VA (53.0 letters versus 66.4, *p* = 0.002) but without a significant difference in VA gain (4.9 letters versus 6.8, *p* = 0.582). Despite a low baseline VA, this subgroup of patients tends to have greater visual gain, encouraging treatment with DEX-implant in such advanced-stage disease. However, some baseline anatomic parameters, such as the presence of EZAs or DRILs, negatively influenced final vision.

## 1. Introduction

Diabetic macular edema (DME) is a major source of visual impairment throughout industrialized countries as the incidence of diabetes dramatically increases worldwide [1]. Pathogenesis and etiology of DME are multifactorial with alteration of the blood-retinal barrier, causing extravasation of fluid into the extracellular space that appears clinically as macular edema, resulting in visual loss [2].

In the past years, new treatments such as intravitreal anti-vascular endothelial growth factor (anti-VEGF) agents have replaced laser photocoagulation and are now considered as the gold standard for treating DME [3,4,5]. However, about 30–35% of patients are non-responsive to anti-VEGF [6] or have had an inadequate response, especially in real-life practice where patients were less treated than pivotal studies [7,8]. Predictive factors of response to treatment are thus required to help the clinician to best adapt the therapy, regardless of the regimen or the therapeutic class used. Dexamethasone implant (DEX-implant) is approved in first-line treatment for DME and anti-VEGF [9]. It has been demonstrated that DEX-implant can achieve similar rates of VA improvement compared to an anti-VEGF for DME, with even better anatomic outcomes and requiring fewer injections [10,11]. As shown for anti-VEGF [12], a recent study by Al-Khersan et al. demonstrated that treatment response to DEX-implant at 3 months is directly correlated to the overall change in best-corrected visual acuity (BCVA), but almost all the patients in this study were non-naïve to anti-VEGF and were considered as suboptimal responders [11].

The purpose of this study was to evaluate the overall VA gain and VA gain profiles between patients with drug-naïve DME treated by DEX-implant. We also assessed the baseline anatomical and functional factors that could be correlated with the response to the treatment in real-life conditions.

## 2. Materials and Methods

### 2.1. Design of the Study

We conducted a multi-center retrospective observational study in 11 retinal centers in France. The study included patients who had drug-naive DME and who had received at least one DEX-implant between 2011 and 2016.

The inclusion criteria were presenting a drug naive DME, without any DME treatment before, except focal laser more than 3 months before. Exclusion criteria were prior cataract surgery less than 6 months before inclusion, idiopathic epiretinal membrane (ERM), or prior vitreous surgery for ERM. Signed informed consent was received from the patients.

A complete ophthalmological examination including with best-corrected visual acuity (BCVA) in Early Treatment Diabetic Retinopathy Study Letters (ETDRS), which is standard of care in these 11 institutions, intraocular pressure (IOP) measurement, slit-lamp biomicroscopy, fundus examination, and spectral-domain optical coherence tomography (SD-OCT) was performed at baseline and at every follow-up visit. In our study, for each patient, retreatment with DEX-implant was indicated if there was a recurrence of DME defined as the presence of visual impairment associated with intraretinal fluid on spectral-domain optical coherence tomography. As long as DEX-implant is whether completely or partially effective on the macular edema, we keep using that treatment and do not switch to another molecule such as anti-VEGF nor to laser treatment. In case of no or insufficient response to DEX-implant, patients were switched to anti-VEGF. The research was conducted in accordance with the Declaration of Helsinki. An international review board approved the study on 23 November 2018 (Ethics Committee of the French Society of Ophthalmology, IRB00008855).

### 2.2. Data Collection

All patients’ data were collected from a retrospective review of medical charts and were identified using an in-house medical database. Patients’ characteristics, past medical history including previous ocular history were recorded. Findings from the ophthalmological examination at each visit were retrospectively collected. Intraocular hypertension was defined as >21 mmHg.

SD-OCT analysis included central subfield macular thickness (CSMT), presence of subretinal fluid (SRF), cystic macular changes, central exudates, hyperreflective foci, alterations of the ellipsoid zone (EZA), and disorganization of retinal inner layers (DRILs) throughout the study period. Hyperreflective foci are well-circumscribed hyperreflective particles that are 20 µm to 40 µm in diameter on SD-OCT and are indicative of an increase in inflammation in the retina. Central exudates are bigger and correspond to the yellow spots on retinographies [13].

### 2.3. Outcome Measures

The main outcome measure was the mean change in BCVA from baseline over the follow-up period. We also identified the VA gain profiles of each patient regarding the baseline characteristics. Secondary endpoints were to assess baseline anatomical prognostic factors of treatment response, measurements of mean change in BCVA, changes in central retinal thickness, the number of injections, and percentage of functional and anatomical responders.

We used the Diabetic Retinopathy Clinical Research network (DRCR-net) definition and thus assessed functional response as a gain of at least 5 letters or more during follow-up and anatomical response as a CSMT decrease ≥20% during follow-up [6].

### 2.4. Statistical Analysis

Categorical variables were described using absolute and relative frequencies, and quantitative variables using mean, standard deviation (+/−SD), and range. Categorical and quantitative variables were compared between groups with the Chi^2^ test and the *t*-test respectively. BCVA and CSMT evolutions were studied with linear mixed-effects models [14] to take into account the within-subject correlation of the repeated observations over time, and the inclusion of patients with a varying number of measurements. BCVA was expressed using the absolute measured value or defined as change from the baseline value. Identifications of BVCA trajectories groups was done with KmL, which is a non-parametric algorithm used for longitudinal data classification [15]—Kml is a non-parametric method for clustering patients into distinct groups based on the evolution (i.e., trajectory) of a longitudinal variable (i.e., a variable measured repeatedly at different time points). The method is able to deal with missing values and allows for different measurements numbers and time-points numbers by patient. Time/visits were entered as an indicator variable in the mixed model, allowing comparisons of VA between groups for each time point. The categories obtained can then be used, for instance, in a regression model, either as predictors or as dependent variables. The algorithm was asked to rank BCVA gain trajectories of patients in two or three groups. Only individual VA trajectories were used to define groups, and this was done independently of baseline VA. Once groups were defined, the association between baseline patients’ characteristics, including baseline VA, and groups was assessed in order to identify characteristics potentially predicting VA gain trajectories. For this secondary analysis, the Chi^2^ test and the *t*-test were used to compare categorical and quantitative variables between groups, respectively. The R software program was used to perform all analyses, and for each test, the 0.05 significance level was used.

## 3. Results

### 3.1. Study Population

Of 107 consecutive patients, 129 eyes with drug-naive DME treated by DEX-implant were included. The baseline characteristics of patients and studied eyes are listed in Table 1. The median follow-up was 13 months with 25% of patients being followed for more than 2 years. The mean number of follow-up visits was 6.0 (±4.9).

### 3.2. Overall Functional and Anatomical Efficacy Description

An improvement of BCVA was observed during the two-year follow-up period. Mean BCVA improved from 54 letters (±18.1) at baseline to 60.4 letters (±18.2) at month 2 (*n* = 97), 60.8 letters (±17.1) at month 8 (*n* = 59), 60.6 letters (±18.8) at 1 year (*n* = 48), and 62.6 letters (±14.5) at 2 years (*n* = 25). Mean BCVA gain was +7.3 letters (±12.6) at 2 months (*n* = 97), +4.9 letters (±13.0) at 8 months (*n* = 59), +4.7 letters (±18.6) at 1 year (*n* = 48). and +7.0 letters (±9.9) at 2 years (*n* = 25). Mean VA gain was statistically significant with an improvement of +4.7 letters at 1 year (*p* = 0.016) and +7 letters at 2 years (*p* = 0.001) (Figure 1). A total of 97 eyes (76%) were functional responders and 84% of the functional response have occurred directly after the first intravitreal injection (IVI). The cumulative proportion of patients who gained ≥15 letters in at least one visit was 14.7% at month 2, 30.2% at month 8, 31.8% at month 12, 33.3% at month 18, 34.9% at month 24, and 38.8% at the end of follow-up.

Mean CSMT decreased from 476.4 μm (±142.6) at baseline (*n* = 123) to 309.8 μm (±83.8) at 2 months (*n* = 93), 381.2 μm (±145.2) at 12 months (*n* = 49), and 358.6 μm (±109.9) at 24 months (*n* = 29) (Figure 2). A total of 88 patients (68%) had a CSMT of 300 μm or less during follow-up, and 82% were anatomical responders.

The mean number of IVI per patient was 2.2 (±1.5) with 78 (59%) patients receiving two or more injections of DEX-implant during the complete follow-up. The average interval of reinjection was 7 months (±3.5); 14% (18 eyes) were switched to another molecule (anti-VEGF) for poor functional response to DEX-implant, as defined as less than 5 ETDRS letters gain, and 8% (10 eyes) stopped being treated because of high IOP (>25 mmHg).

### 3.3. Analyzes of VA Gain Profiles

The Kml algorithm presented the best classification in two groups. Group A was composed of 96 patients (74%) whose mean VA gain was low on all follow-up, and Group B composed of 33 patients (26%) whose VA gain 20 letters from month 2 and remained high at 2 years.

The mean VA gain in Group A was +2.3 letters (range −0.1–4.6) at 2 months, −3.0 letters (range −6.1–0.2) at 1 year, and +2.3 letters (range −1.3–6.0) at 2 years. In Group B, the mean VA gain was +19.2 letters (range 15.5–22.8) at 2 months, +21.3 letters (range 15.9–25.9) at 1 year, and +18.6 letters (range 11.0–23.8) at 2 years. The difference between the two groups was statistically significant from month 2 to the end of the follow-up study (*p* < 0.001).

While comparing the characteristics of patients between the two groups, only the low baseline VA appeared to be significantly associated with Group B (Table 2). There were only 11 patients (11.3%) in Group A with a VA at baseline <37 letters versus 12 (34.3%) patients in Group B (*p* < 0.001) and 27 (27.8%) patients in Group A with VA at baseline ≥69 letters versus one (2.9%) patient in Group B (*p* < 0.001). Concerning the 11 patients in Group A with VA <37 letters at baseline, they had more frequent DRILs (*p* = 0.001), EZAs (*p* < 0.001), and macular ischemia (*p* = 0.004) seen on Optical Coherence Tomography (OCT) and Fluorescein Angiography (FA) than patients with VA ≥37 letters.

The probability of belonging to Group B was significantly higher in patients with low baseline VA <37 letters (*p* < 0.001) and between 37–68 letters (*p* = 0.017). There was no other factor that stood out as significantly associated. In particular, there was no significant difference between the two groups regarding the numbers of patients undergoing cataract surgery: eight phakic eyes (40%) of Group B were operated on, in comparison to 15 eyes (37.5%) of Group A (*p* = 0.99). Figure 3 illustrates changes in final VA and VA gain depending on baseline VA and shows that the lower baseline VA, the lower the final VA at 24 months. On the contrary, the lower the baseline VA, the greater the VA gain.

### 3.4. Anatomical Prognosis Factor for VA Gain

Subgroup analysis of prognostic factors revealed that improvement in BCVA was not affected by age, sex, blood pressure nor baseline CSMT. There was no statistical difference in VA changes regardless of lens status, presence of SRF, central exudates, or hyperreflective foci (Table 3). However, subgroup analysis highlighted a significant statistical difference in BCVA changes in eyes with or without ellipsoid zone alterations (EZAs) (*p* < 0.001), in eyes with or without disorganization of retinal inner layers (DRILs) (*p* < 0.001), or both (*p* = 0.008). EZAs were associated with a lower final VA (47.5 letters versus 66.6 letters, *p* < 0.001). The presence of DRILs was also associated with a lower final VA (46.8 letters versus 64.7 letters, *p* = 0.002). In the same way, a combination of both EZAs and/or DRILs was associated with a lower final VA (53.20 letters versus 66.40 letters, *p* = 0.002), but there was no significant difference in VA gain (4.94 letters versus 6.81 letters, *p* = 0.582).

### 3.5. Safety

A total of 104 (79%) eyes had an IOP ≤21 mmHg at any visit during the follow-up study and only 10 eyes (7.4%) had an IOP ≥25 mmHg. A total of 39 patients (29%) needed topical antiglaucoma medication and all of them were controlled with topical monotherapy [16]. No glaucoma filtering surgery was performed in the present study. Cataract surgery was performed in 42.6% of the phakic eyes (*n* = 26/61) during the study period, and 65% of these surgeries occurred after the first IVI. There was no significant difference between the two groups regarding the numbers of patients undergoing cataract surgery—eight phakic eyes (40%) of Group B were operated on, in comparison to 15 eyes (37.5%) of Group A (*p* = 0.99). The improvement in BCVA after cataract surgery could be due to the reduction of macular edema (ME) following DEX-implant injection and cannot be differentiated from cataract extraction. Figure 4 shows the effects of cataracts on BCVA over time. The mean improvement in BCVA obtained using DEX-implant in the subgroup of phakic patients having undergone cataract surgery improved over time.

No other serious ocular or systemic adverse events were observed throughout the follow-up.

## 4. Discussion

Our study described two VA gain profiles after treatment by DEX-implant. One group whose average VA gain was less than five letters (Group A) and another group with an average gain of 20 letters (Group B). The Kml method, used herein, allows us to overcome the use of arbitrary definitions of thresholds, as the groups are defined without making any a priori hypothesis on the form of the BCVA evolution. The probability of belonging to the second group was significantly higher in patients with lower baseline VA. We found no other factors associated with better VA gain, although an increased retinal thickness is close to the significance. These findings are consistent with data on anti-VEGF studies, which have shown that patients with low baseline VA gained more than patients with higher baseline VA [17]. The association between worse baseline VA and a better VA improvement may be affected by a “ceiling effect” on the degree of improvement possible for those with better baseline VA [18]. Nevertheless, these patients have a lower final VA, which is an argument to treat the patients early, when VA is still high enough, even if the expected VA gain is less important. The lower final VA in patients with low baseline VA might be explained by anatomical retinal scars and a definite loss of photoreceptors that do not recover or regenerate after edema regression. Thus, an additional aim of our study was to describe anatomical biomarkers that could predict final VA and VA gain. EZAs and presence of DRILs were associated with a lower final VA but interestingly without any difference in VA gain. DRILs were already known as an effective non-invasive biomarker of future VA response in eyes with DME, in addition to EZAs [19,20,21,22,23]. Lens status, subretinal fluid (SRF), central exudates, and hyperreflective foci were not significantly associated with a significant difference in VA at 2 years in our study, although the results for the hyperreflective foci were contradictory with another recent study [20].

Bressler et al. described anatomical predictive factors for response to anti-VEGF therapy and found that baseline CSMT was the strongest predictor of anatomic outcome. They also showed that reduction in CSMT during the first year of treatment was associated with better VA outcomes [18]. More recently, it was shown, regarding DME treated by ranibizumab, that the height of intraretinal cystoid fluid spaces at baseline was a better predictor of functional and anatomical improvement than the central retinal thickness alone [20].

Patients included in this real-life study were drug-naive, knowing that the use of previous treatment for DME is a negative factor of recovery [24]. Herein, DEX-implant was chosen in first-line therapy as it can decrease the burden of the treatment injection for diabetic patients that often have many other multidisciplinary appointments, keeping a high efficiency and low rates of non-responders [9]. Moreover, the absence of systemic side effects such as cardiovascular events was another priority to choose this molecule because the presence of DME is a predictor of cardiovascular morbidity and mortality [25].

Regarding the overall population included, our study confirmed the good anatomical, functional efficacy, and safety of DEX-implant for DME [26,27,28]. However, consistently with literature, a deterioration of BCVA was noticed at the 12-month visit, which was probably caused by a steroid-induced cataract, with subsequent improvement after removal [29]. In parallel with these functional responses, anatomical outcomes showed a marked decrease in CSMT that remained low all over the study follow-up. However, these changes did not correlate with changes in BCVA in our regression analysis, showing that other anatomical biomarkers should play a role in the functional response [11]. It is also important to highlight that these anatomical and functional results were obtained with a low number of injections in this real-life study.

Another interesting finding in this study is that the VA gain outcomes after the first DEX-implant seemed to predict the final BCVA changes over 2 years. These results are consistent with a recent study of Al-Khersan et al. who showed that early treatment response (at 3 months) to DEX-implant is directly correlated with the overall change in BCVA [11]. However, these patients were not drug naive for DME, more than 90% having received previous anti-VEGF IVI, without any details of a real washout period. It is well known that previous treatment of DME is a negative factor of recovery and this could underestimate the final VA gain [24]. These results with DEX-implant are similar to those from anti-VEGF [12,30].

Our study has some limitations. Its retrospective design limits the exhaustiveness of data collection. Only a few numbers of the patients included were followed up until year 2 and results regarding that time point must be taken with caution. Indeed, long-term treatment by DEX-implant could be given only in patients with acceptable response to the molecule. However, only 14% of patients were switched to another therapy during the study, showing that only a few numbers of patients poorly respond to DEX-implant. Moreover, the Kml method takes into account the imbalance of follow-up visits and associated missing values. Even though this study included the largest number of drug-naive patients treated by DEX-implant, it did not allow to identify an intermediate group for VA gain profiles. However, despite the aforementioned limitations of retrospective analysis, the present work was performed with data gathered from patients in the real-world clinical setting rather than from selective clinical trials with thoroughly monitored patient follow-up, which is unlikely to occur in the clinical setting. Thus, despite its limitations, the present data are more likely generalizable to the clinical setting.

## 5. Conclusions

The main parameters identified for better VA gain appear to be dependent on the baseline VA. Moreover, we identified baseline DRIL or EZA as poor prognosis factors for final VA at 2 years. Further studies are needed to highlight new biomarkers associated with treatment response, allowing the clinicians to personalize the therapeutic options.

## Figures and Tables

**Figure 1 pharmaceutics-13-00194-f001:**
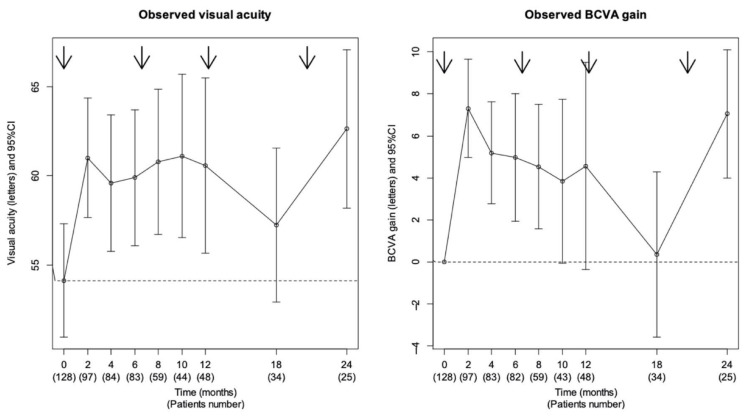
Functional efficacy. Mean best-corrected visual acuity (BCVA) and BCVA gain in Early Treatment Diabetic Retinopathy Study Letters (ETDRS) from baseline during the follow-up. When the vertical line cuts the dotted line, the value is not statistically different from the baseline value. Arrows represent the median dates of injection.

**Figure 2 pharmaceutics-13-00194-f002:**
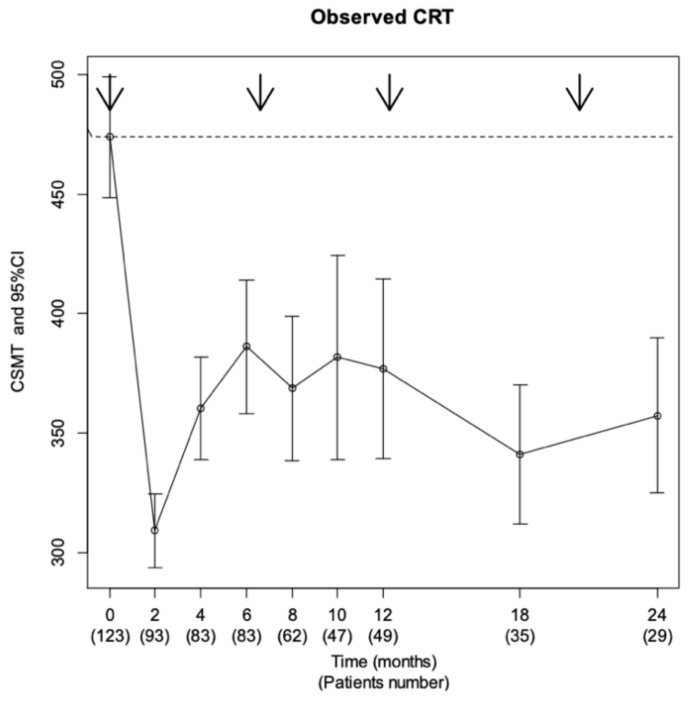
Anatomical efficacy. Evolution of mean central subfield macular thickness (CSMT) during the study period. Arrows represent the median dates of injection.

**Figure 3 pharmaceutics-13-00194-f003:**
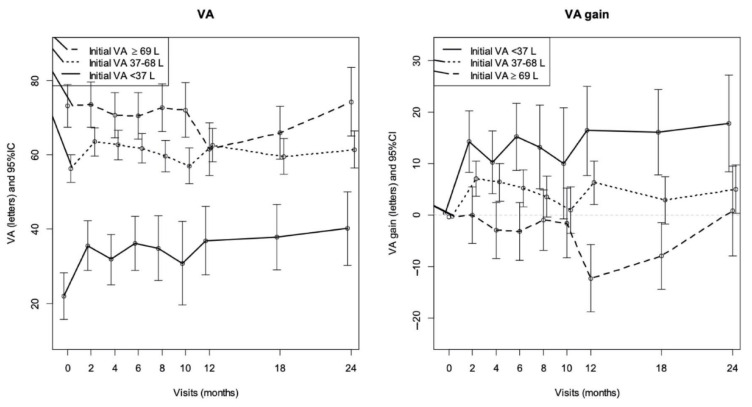
Changes in final VA and VA gain depending on baseline VA.

**Figure 4 pharmaceutics-13-00194-f004:**
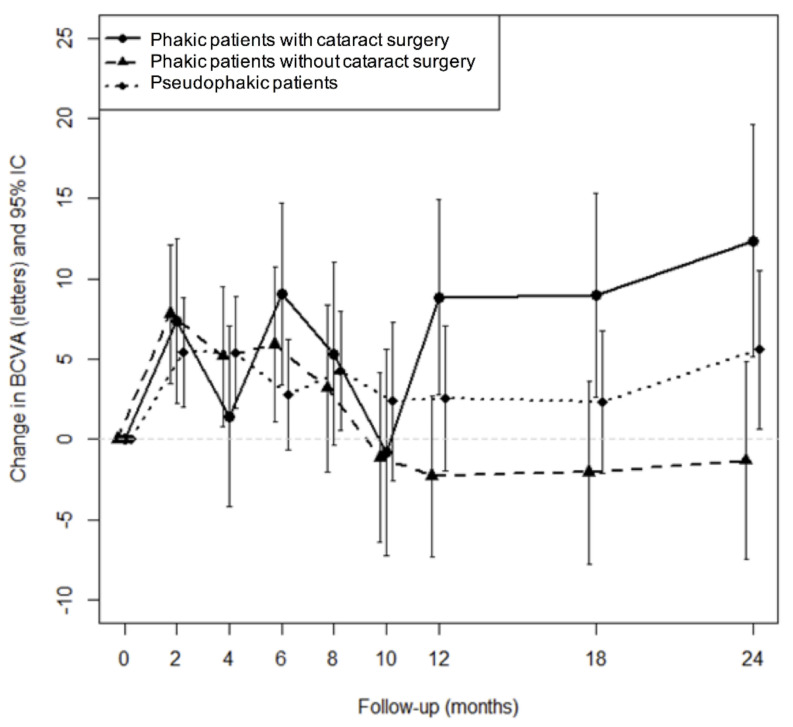
Treatment efficacy according to the phakic status of the patient.

**Table 1 pharmaceutics-13-00194-t001:** Baseline characteristics of patients and study eyes. (BCVA = best-corrected visual acuity; CSMT = central subfield macular thickness; HIP = high intraocular pressure).

Characteristics		
Mean Age, Years (Range)	65.8 Years (34.9–86.1)	
Mean follow-up, months (±SD)	16.5 (±11.4)	
Mean BCVA, ETDRS letters (±SD)	54.0 (±18.1)	
Mean subfield central macular thickness (CSMT), μm (±SD)	476.4 (±142.6)	
Mean IOP, mmHg (±SD)	14.5 (±3.0)	
Eye	Right	Left
	54 (42%)	75 (58%)
Sex	F	M
	67 (52%)	62 (48%)
Lens status	Phakic	Pseudophakic
	60 (46%)	69 (54%)
Diabetes	Type 1	Type 2
	13 (10%)	116 (90%)
	Number of Patients with (%)	
High blood pressure	94 (73%)	
Glaucoma/HIP	6 (4.5%)	
Vitrectomy	5 (4%)	
Panretinal photocoagulation (PRP)	73 (56%)	
Focal laser	26 (21%)	
Subretinal Fluid (SRF)	15 (12%)	
Central Exudates	14 (11%)	
Hyperreflective foci	44 (34%)	
Ellipsoid Zone Alterations (EZA)	40 (31%)	
Disorganization of Retinal Inner Layers (DRIL)	41 (32%)	
DRIL or EZA	58 (45%)	

**Table 2 pharmaceutics-13-00194-t002:** Patient baseline characteristics between Group A and Group B for visual acuity (VA) gain profiles. (BCVA = best-corrected visual acuity; CSMT = central subfield macular thickness; HIP = high intraocular pressure; HBP = high blood pressure; PRP = panretinal photocoagulation; SRF = subretinal fluid; EZAs = ellipsoid zone alterations; DRILs = disorganization of retinal inner layers).

Observations	Group A	Group B	*p*-Value
*n*	96	33	
Age (+/−SD)	66.1 (+/−10.8)	64.1 (+/−12.6)	0.428
Baseline BCVA (+/−SD)	57.9 (+/−16.7)	43.4 (+/−19.0)	0.001
Baseline CSMT (+/−SD)	455.8 (+/−133.6)	521.2 (+/−161.2)	0.055
Sex			0.835
Female	50 (52.6%)	16 (48.5%)	
Male	45 (47.4%)	17 (51.5%)	
Lens status			0.103
Phakic	40 (42.1%)	20 (60.6%)	
Pseudophakic	55 (57.9%)	13 (39.4%)	
Diabetes			0.921
Type 1	9 (9.5%)	4 (12.1%)	
Type 2	86 (90.5%)	29 (87.9%)	
HBP	72 (75.8%)	21 (63.6%)	0.262
PRP	55 (57.9%)	18 (54.5%)	0.896
Focal laser	20 (21.1%)	6 (18.2%)	0.919
SRF	10 (10.5%)	4 (12.1%)	1
Central Exudates	10 (10.5%)	4 (12.1%)	1
Hyperreflective foci	34 (35.8%)	9 (27.3%)	0.497
EZAs	33 (34.7%)	7 (21.2%)	0.22
DRILs	31 (32.6%)	9 (27.3%)	0.723
DRILs or EZAs	46 (48.4%)	11 (33.3%)	0.194

**Table 3 pharmaceutics-13-00194-t003:** Anatomical prognosis factor for final visual acuity (VA) and VA gain at 24 months. (BCVA = best-corrected visual acuity; CSMT = central subfield macular thickness; HIP = high intraocular pressure; HBP = high blood pressure; PRP = panretinal photocoagulation; SRF = subretinal fluid; EZAs = ellipsoid zone alterations; DRILs = disorganization of retinal inner layers).

Observations at 24 Months	Final VA (L)			VA Gain (L)		
Baseline BCVA	**<37 L**	**>69 L**	***p*-value**	**<37 L**	**>69 L**	***p*-value**
	39.8	75.1	<0.001	17.3	1.5	0.02
	**Yes**	**No**	***p*-value**	**Yes**	**No**	***p*-value**
Pseudophakic	58.3	62.5	0.39	7.15	5.3	0.52
Panretinal photocoagulation (PRP)	56.5	61.5	0.22	3.6	7.5	0.25
Subretinal Fluid (SRF)	60.8	50.8	0.12	6.6	2.64	0.40
Central Exudates	59.8	56.7	0.88	6.2	4.7	0.70
Hyperreflective foci	62.7	52.5	0.05	8.1	2.1	0.23
Ellipsoid Zone Alterations (EZAs)	66.6	47.5	<0.001	7.5	2.9	0.22
Disorganization of Retinal Inner Layers (DRILs)	64.7	46.8	0.002	6.0	5.3	0.85
DRILs or EZAs	66.4	53.2	0.002	6.8	4.9	0.58

## Data Availability

Data are available upon request to the corresponding author.

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
