# Peer review of "Visual Acuity Gain Profiles and Anatomical Prognosis Factors in Patients with Drug-Naive Diabetic Macular Edema Treated with Dexamethasone Implant: The NAVEDEX Study"

_pharmaceutics, 2021, doi:10.3390/pharmaceutics13020194_

Round 1
Reviewer 1 Report
I congratulate the Authors for having evaluated, in detail even if the number of patients is not very high, through a multicenter study carried out, independently and separately in 11 retinal centers in France, this represents a strength of the study.
The study is well conducted, also the statistical evaluation of safety and efficacy.
This study reinforces the results of other spontaneous studies that deserve to be mentioned.
I therefore recommend:
on page2, Line n. 11 … less treated than pivotal studies (Holz et al. 2013; Kodjikian et al. 2018)…. the refernce:
Pacella E, La Torre G, Impallara D, Malarska K, Turchetti P, Brillante C, Smaldone G, De Paolis G, Muscella R, Pacella F. Efficacy and safety of the intravitreal treatment of diabetic macular edema with pegaptanib: a 12-month follow-up. Clin Ter. 2013;164(2):e121-6. doi: 10.7417/CT.2013.1543. PMID: 23698213.
RESULTS
Safety
on page 9, …..medication and all of them were controlled with a topical monotherapy…...addition the references
Pacella E, Loffredo L, Malvasi M, Trovato Battagliola E, Messineo D, Pacella F, Arrico L. Effects of Repeated Intravitreal Injections of Dexamethasone Implants on Intraocular Pressure: A 4-Year Study. Clin Ophthalmol. 2020 Oct 29;14:3611-3617. doi: 10.2147/OPTH.S265691. PMID: 33154620; PMCID: PMC7605966.
Discussion
on page11 , ………………..These results with DEX-implant are similar to those from anti-VEGF as shown by Gonzalez et al (Gonzalez et al. 2016)………….
Pacella F , Turchetti P, Piraino D. C. et al .Evaluation of retinal ganglion cell layer in patients with macular edema and treated with intravitreal anti-VEFG and corticosteroid Senses Sci 2019: 6 (2) 767-780 doi: 10.14616/sands- 2019-6-767780
.
Author Response
Reviewer Comments:
Authors: We thank the reviewer for their comments and suggestions to improve our manuscript. Please find below a point-by-point reply to the comments made.
Reviewer #1:
*I congratulate the Authors for having evaluated, in detail even if the number of patients is not very high, through a multicenter study carried out, independently and separately in 11 retinal centers in France, this represents a strength of the study. The study is well conducted, also the statistical evaluation of safety and efficacy. This study reinforces the results of other spontaneous studies that deserve to be mentioned.
Authors: Thank you for such comments
*I therefore recommend on page2, Line n. 11 … less treated than pivotal studies (Kodjikian et al. 2018) …. the reference: Pacella E, La Torre G, Impallara D, Malarska K, Turchetti P, Brillante C, Smaldone G, De Paolis G, Muscella R, Pacella F. Efficacy and safety of the intravitreal treatment of diabetic macular edema with pegaptanib: a 12-month follow-up. Clin Ter. 2013;164(2): e121-6. doi: 10.7417/CT.2013.1543. PMID: 23698213.
Authors: We added the suggested reference on the manuscript.
*RESULTS - Safety
on page 9, …. medication and all of them were controlled with a topical monotherapy…...addition the references Pacella E, Loffredo L, Malvasi M, Trovato Battagliola E, Messineo D, Pacella F, Arrico L. Effects of Repeated Intravitreal Injections of Dexamethasone Implants on Intraocular Pressure: A 4-Year Study. Clin Ophthalmol. 2020 Oct 29;14:3611-3617. doi: 10.2147/OPTH.S265691. PMID: 33154620; PMCID:PMC7605966.
Authors: We added the suggested reference on the manuscript.
*Discussion
on page11, …. These results with DEX-implant are similar to those from anti-VEGF as shown by Gonzalez et al (Gonzalez et al. 2016) …Pacella F, Turchetti P, Piraino D. C. et al. Evaluation of retinal ganglion cell layer in patients with macular edema and treated with intravitreal anti-VEFG and corticosteroid Senses Sci 2019: 6 (2) 767-780 doi: 10.14616/sands- 2019-6-767780
Authors: We added the suggested reference on the manuscript
Reviewer 2 Report
Brief Overview: Pinto M et al. conducted a retrospective, multi-center, chart review on patients whose eyes were drug naïve diabetic retinopathy treated with DEX implant in order to evaluate the visual acuity (VA) profiles and to assess factors that could influence responses to the treatment. The main finding was that patients with lower baseline VA tended to have a higher average gain of 20 letters compared to those with higher baseline VA.
Overall Comments: The manuscript is well-written and addresses an important problem. The objectives are clearly defined, but no working hypothesis is stated. The following clarifications are needed:
- Abstract:
- Spell out acronyms at first mention.
- The conclusions should be restated to give a take-home message and not rehash the results. For example, what is the significance of low baseline VA having better VA outcomes with this treatment?
- Correct the grammatical error on the last line.
- Introduction:
- Spell out abbreviations at first mention.
- State the working hypothesis.
- Methods:
- State that signed informed consent was received from the patients.
- Based on the study design, one group of patients were administered the DEX implant and followed for VA outcomes. The data was then analyzed to compare baseline versus post treatment over time. Examination of Figures 1 and 2 suggests that comparisons were made over 8 time intervals to baseline. The statistical analyses for this is not t-test, but ANOVA for repeated measurements because analysis is done pre and post within the same group.
- Also, if change in BCVA from baseline is the main the outcome (page 3), then percentage change from baseline should be employed.
- Based on the design, the authors conducted a sub-analysis (stratification) and divided the data between two groups of patients: 1) who had an average VA gain of 5 letters (A); and 2) patients with an average VA gain of 20 letters (B). The statistical analysis for this is unpaired t-test. This should be clarified in the methods and statistical analyses sections to avoid unnecessary confusion.
- Results:
- Page 3, the median follow up was 13 months, but only 25% was followed for 2 years. State clearly when the final VA and VA gain were determined and how many eyes were involved.
- Table 2: Clarify in the table legend that the data is baseline characteristics.
- Table 3: Clarify the groups (A and B).
- Discussion:
- The statements on page 10 appear contradictory. On lines 3 and 4, the authors state: “the probability of belonging to the second group was significantly higher in patients with lower baseline VA”. This implies that patients who had lower baseline tended to have better VA outcomes. Lines 6 and 7 states: “These findings are consistent with data on anti-VEGF studies which have shown that patients with low baseline VA gained more than patients with higher baseline VA”. Also confirming that patients who had lower baseline VA tended to have better VA outcomes. However, lines 9 and 10 states: “The lower final VA in patients with low baseline VA might be explained by anatomical retinal scars…”. This sentence must be clarified because this is the major finding of the study (see conclusions statement).
- Once the statement is clarified, more discussion is needed to explain this phenomenon, i.e. why did the patients with higher baseline VA have lower VA gains post treatment, or vice versa.
Author Response
Reviewer #2:
Brief Overview: Pinto M et al. conducted a retrospective, multi-center, chart review on patients whose eyes were drug naïve diabetic retinopathy treated with DEX implant in order to evaluate the visual acuity (VA) profiles and to assess factors that could influence responses to the treatment. The main finding was that patients with lower baseline VA tended to have a higher average gain of 20 letters compared to those with higher baseline VA.
Overall Comments: The manuscript is well-written and addresses an important problem. The objectives are clearly defined, but no working hypothesis is stated. The following clarifications are needed:
- Abstract:
- Spell out acronyms at first mention.
Authors: We have now spelled out all of the acronyms in the abstract section as well as in the whole manuscript.
- The conclusions should be restated to give a take-home message and not rehash the results. For example, what is the significance of low baseline VA having better VA outcomes with this treatment?
Authors: we restated our conclusion as suggested. Page 1-2, line 45 to 47: Despite a low baseline VA, this subgroup of patients tends to have the greater visual gain, encouraging a treatment with DEX-implant in such advanced stage disease. However, some baseline anatomic parameters, the presence of EZA or DRIL, negatively influenced final vision.
- Correct the grammatical error on the last line.
Authors: We have modified the last sentence accordingly to the comment 2.
- Introduction:
- Spell out abbreviations at first mention.
Authors: We have now spelled out all of the abbreviations.
- State the working hypothesis.
Authors: Our working hypothesis is such as written page2 line 73 to 76: “The purpose of this study was to evaluate the overall VA gain and VA gain profiles between patients with drug-naïve DME treated by DEX-implant. We also assessed the baseline anatomical and functional factors that could be correlated with the response to the treatment in real-life conditions”.
- Methods:
- State that signed informed consent was received from the patients.
Authors: We have now stated that all patients received and signed an informed consent in accordance to the recommendation of the ethical committee recommendation, page 2 line 85.
- Based on the study design, one group of patients were administered the DEX implant and followed for VA outcomes. The data was then analyzed to compare baseline versus post treatment over time. Examination of Figures 1 and 2 suggests that comparisons were made over 8-time intervals to baseline. The statistical analyses for this is not t-test, but ANOVA for repeated measurements because analysis is done pre and post within the same group.
Authors: We have included patients with drug-naive DME that undergo a treatment by DEX-implant. Thus, we have one group of patients that we followed for VA and anatomical outcomes. At each visit, VA and CRT parameters were collected that allowed to construct the Figures 1 and 2. However, as stated in the statistical analysis section, BCVA and CSMT evolutions were studied with linear mixed-effects models.
- Also, if change in BCVA from baseline is the main the outcome (page 3), then percentage change from baseline should be employed.
Authors: We chose to use the mean change in BCVA from baseline as the main outcome measure to allow comparison between our study and others in the literature. In fact, mean change in BCVA is the principal outcome measure in many pivotal randomized-controlled trials but also in observational retrospective real-world studies. If the reviewer accept, we prefer to keep mean change in BCVA as the main outcome measure.
- Based on the design, the authors conducted a sub-analysis (stratification) and divided the data between two groups of patients: 1) who had an average VA gain of 5 letters (A); and 2) patients with an average VA gain of 20 letters (B). The statistical analysis for this is unpaired t-test. This should be clarified in the methods and statistical analyses sections to avoid unnecessary confusion.
Authors: We agree with the reviewer's comment and clarified this issue in the Statistical Analysis section. Page 3 line 135-136: “For this secondary analysis, the Chi² test and the t-test were used to compare categorical and quantitative variables between groups, respectively”.
- Results:
- Page 3, the median follow-up was 13 months, but only 25% was followed for 2 years. State clearly when the final VA and VA gain were determined and how many eyes were involved.
Authors: we have included the total number of participants at each timepoints in the figure 1 and 2 below the x-axis. To improve the readability, we have included this number in the text. Page 4 -5, line 149 to 163.
- Table 2: Clarify in the table legend that the data is baseline characteristics.
Authors: we added the word “baseline characteristics” in the legend of table 2, page 7 line 194.
- Table 3: Clarify the groups (A and B).
Authors: In the table 3, we did not analyzed anatomical prognosis factors depending on groups A and B, but for all the patient. As such, we cannot clarify the groups A and B.
- Discussion:
- The statements on page 10 appear contradictory. On lines 3 and 4, the authors state: “the probability of belonging to the second group was significantly higher in patients with lower baseline VA”. This implies that patients who had lower baseline tended to have better VA outcomes. Lines 6 and 7 states: “These findings are consistent with data on anti-VEGF studies which have shown that patients with low baseline VA gained more than patients with higher baseline VA”. Also confirming that patients who had lower baseline VA tended to have better VA outcomes. However, lines 9 and 10 states: “The lower final VA in patients with low baseline VA might be explained by anatomical retinal scars…”. This sentence must be clarified because this is the major finding of the study (see conclusions statement).
Authors: According to the reviewer suggestion we have clarified those sentences. Page 10, line 238 to 246: “The Kml method, used herein, allows to overcome the use of arbitrary definitions of thresholds, as the groups are defined without making any a priori hypothesis on the form of the BCVA evolution. The probability of belonging to the second group was significantly higher in patients with lower baseline VA. We found no other factors associated with better VA gain, although an increased retinal thickness is close to the significance. These findings are consistent with data on anti-VEGF studies, which have shown that patients with low baseline VA gained more than patients with higher baseline VA (Dugel et al. 2016). The association between worse baseline VA and a better VA improvement may be affected by a “ceiling effect” on the degree of improvement possible for those with better baseline VA (Bressler SB et al.)”.
- Once the statement is clarified, more discussion is needed to explain this phenomenon, i.e., why did the patients with higher baseline VA have lower VA gains post treatment, or vice versa.
Authors: We have added more discussion about this phenomenon, describing a probable ceiling effect for patients with better baseline VA. Page 10, line 244 to 246: “The association between worse baseline VA and a better VA improvement may be affected by a “ceiling effect” on the degree of improvement possible for those with better baseline VA (Bressler SB et al.)”.
This manuscript is a resubmission of an earlier submission. The following is a list of the peer review reports and author responses from that submission.